# FetchBench: A Simulation Benchmark for Robot Fetching

**Beining Han**[*]    **Meenal Parakh**    **Derek Geng**
**Jack A Defay**    **Gan Luyang**    **Jia Deng**

Princeton University

**Abstract:** Fetching, which includes approaching, grasping, and retrieving, is a critical challenge for robot manipulation tasks. Existing methods primarily focus on table-top scenarios, which do not adequately capture the complexities of environments where both grasping and planning are essential. To address this gap, we propose a new benchmark FetchBench, featuring diverse procedural scenes that integrate both grasping and motion planning challenges. Additionally, Fetch-Bench includes a data generation pipeline that collects successful fetch trajectories for use in imitation learning methods. We implement multiple baselines from the traditional sense-plan-act pipeline to end-to-end behavior models. Our empirical analysis reveals that these methods achieve a maximum success rate of only 20%, indicating substantial room for improvement. Additionally, we identify key bottlenecks within the sense-plan-act pipeline and make recommendations based on the systematic analysis. The code for the benchmark is available at https://github.com/princeton-vl/FetchBench-CORL2024 .

**Keywords:** Grasping; Benchmark; Imitation Learning

## 1 Introduction

We study robotic fetching: the full process of approaching, grasping, and retrieving an object from the environment. Fetching unknown objects in *unseen* environments is crucial to various manipulation tasks and applications. Yet, current approaches [1, 2, 3, 4, 5] primarily focus on computing grasp poses for unknown objects in *limited environment* settings, such as table-top and bins. This contrasts with everyday scenarios where we often need to pick objects from shelves, cabinets, drawers, and other locations. In these situations, the task extends beyond just finding suitable grasp poses. It also involves moving objects from cluttered environments to a desired location, e.g., the free space near the robot for in-hand manipulation [6] or above a container to drop the object [7].

Given the gap between the complexities of real-world scenarios and those typically addressed in current research, it remains unclear *whether existing fetching methods can handle unknown objects in novel and complex daily environments* and what are the failure or edge cases of such methods.

To address these questions, we introduce FetchBench, a benchmark designed for the large-scale evaluation of methods that integrate grasp prediction with motion planning. FetchBench specifically targets the task of grasping and retrieving objects in complex scenarios commonly encountered in daily life, such as retrieving items from shelves, drawers, and cupboards. This contrasts with existing manipulation benchmarks (listed in Table 1), which predominantly focus on general skill learning and have scenarios limited to picking objects from tabletop environments.

We use procedural scenes to generate fetching queries, which are unlimited and comprehensive (Figure 1). In contrast, all other simulation benchmarks use hand-crafted environments and thus are

---

[*]Corresponding author: bh7032@princeton.edu

8th Conference on Robot Learning (CoRL 2024), Munich, Germany.

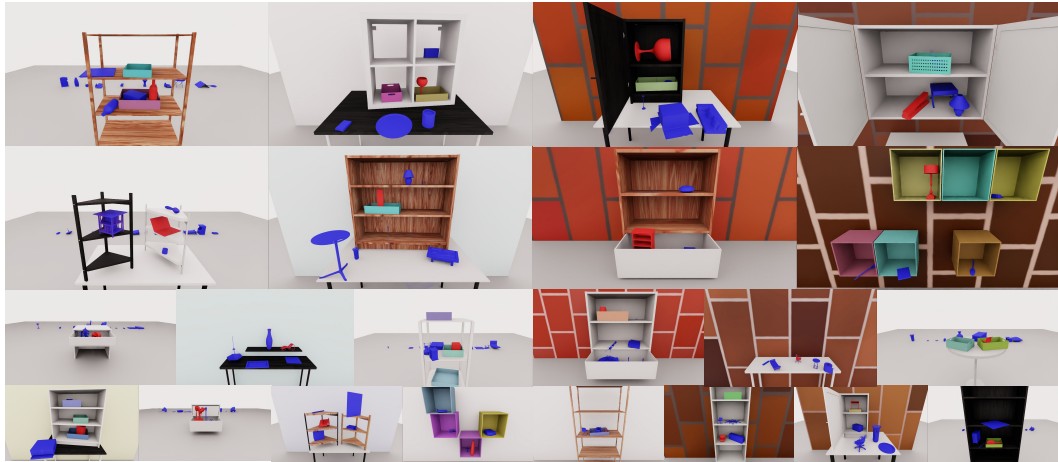

Figure 1: Examples of the fetching task in our benchmark. The red object is the target object that needs to be retrieved from the cluster to the free space in each task. Our scenes and tasks are generated with procedural rules, which mimic daily environments like shelves, cabinets, drawers, baskets, etc. The images are rendered with Isaac-Sim [8].

limited in the variation of the spatial constraint of the cluttered scene. Compared to Cabinet [9], whose procedural assets are not publicly available, we build our scenes with public Infinigen [10] assets and have an increased diversity of in-basket objects.

Based on the benchmark, we implement and evaluate several baselines for the task. In particular, the common pipeline used in previous works [1, 11, 5] is to use grasp pose prediction models to propose candidate grasp poses and then use motion planning [12] algorithms to search for the collision-free motion to reach the target location. We find in Section 6 that this pipeline is no longer reliable when fetching from novel, complex, and partially-observed environments, achieving only 13% success rate. We analyze the failure cases by placing oracle components within the pipeline and point out several challenges that remain on existing motion planning algorithms and grasp pose prediction models (Section 6.2).

In addition to the common pipeline, we opt to learn a large-scale behavior model for the task. We generate a dataset using the procedural scenes and expert grasping demos with CuRobo [13] and ground truth grasp annotations. We find that the best performance comes with combining the common pipeline with the behavior model (Section 6.1): using the conventional pipeline in the approaching phase while using the end-to-end behavior model for moving the grasped object to free-space.

In summary, we make the following contributions:

- FetchBench: A benchmark designed to evaluate robotic methods for fetching unknown objects in unseen environments.
- An empirical study conducted on the benchmark, which identifies key challenges and failure cases of existing methods.
- A dataset generator developed for imitation learning methods in robot fetching tasks.

## 2   Related Works

**Manipulation Benchmark in Simulators.** Benchmarking robotic systems and algorithms in simulation has the advantage of high efficiency, low cost, and reproducibility. Compared to existing benchmarks [14, 17, 31, 22, 18, 24, 15, 16, 27, 29, 21] for robot manipulation (Table 1), our benchmark is unique in the following aspects: First, we specialize in evaluating robot fetching of unknown objects from novel environments, while other benchmarks either focus on general-purpose robot learning algorithms, e.g. imitation learning [22, 18, 17, 27], reinforcement learning [18, 14, 24, 16, 17, 29], or on long-horizon embodied AI challenge by simplifying the grasping pro-

Table 1: Comparison with other robot manipulation benchmarks in simulation. Some entries are adapted from [14]. For rows of [14, 15, 16], the number of scenes denotes the number of rooms. *IL*: Imitation learning. *RL*: Reinforcement learning. *LfD*: Learning from demonstration. *EAI*: Embodied AI. *SPA*: Sense-Plan-Act [17, 15]. *BRL*: Batch RL/Offline RL. *EAI-G*: Embodied AI with abstraction of grasping process. *M2RL*: Meta and Multi-task RL.

| | Platform | Focus | Scenes | Objects | Demos | Baselines |
|---|---|---|---|---|---|---|
| Robosuite [18, 19] | Mujoco [20] | LfD,RL | 1 | 10 | ✓ | IL,BRL,RL |
| RoboCasa [21] | Robosuite [18] | LfD | 120 | 2509 | ✓ | IL |
| RLBench [22] | CoppeliaSim [23] | LfD,RL | 1 | 28 | ✓ | IL,RL |
| Metaworld [24] | Mujoco [20] | M2RL | 1 | 80 | | M2RL |
| Behavior-100 [16] | iGibson2 [25] | EAI | 100 | 1.2k | | RL |
| Maniskill2 [17] | Sapien [26] | LfD,RL | 1 | 2k+ | ✓ | SPA,IL,RL |
| FurnitureSim [27] | Isaac-Gym [28] | LfD | 1 | 35 | ✓ | IL,BRL |
| HumanoidBench [29] | Mujoco [20] | RL | 15 | 20+ | | RL |
| Behavior-1k [14] | OmniGibson [14] | EAI | 306 | 5.2k+ | | RL |
| Habitat2-HAB [15] | Habitat [30] | EAI-G | 6 | 100+ | | SPA,RL |
| ManipulaTHOR [31] | AI2-Thor [32] | EAI-G | 30 | 150 | | RL |
| FetchBench (Ours) | Isaac-Gym [28] | Fetching | Proc | 5.5k+ | ✓ | SPA,IL |

cess [15, 31]. Second, our benchmark provides abundant procedural environments for training and evaluation, with a total of 5.5k objects for grasping. Other benchmarks are limited by the number of artist-designed environments and the number of objects to be grasped.

**Robotic Fetching.** Robotic fetching (grasping) is a popular and challenging problem [33]. Researchers have built robotics systems that can grasp diverse and unknown objects from table-tops [1, 34] and bins [5, 4]. In these works, the common pipeline is to predict grasp poses [35, 3, 2, 34] and use motion planning to approach the grasp pose and retrieve the object out of the cluttered scene [13, 12, 9]. The alternative is to learn end-to-end policy with imitation learning [36, 37]. However, the evaluations in these works, though real-world, are severely limited to ∼10 scenes and ≤100 objects [1, 3, 33] or only focus on collision detection and motion planning while ignoring the grasping [9]. In contrast, FetchBench evaluates the full pipeline study both the modular approaches built around the common pipeline and end-to-end approaches, ablating over different components with oracles. Previous works have not conducted such a study.

## 3    Task Setup

We procedurally generate scenarios (Section 4) in IssacGym [28] and spawn a 7-DOF Franka robot arm equipped with a Franka gripper in front of the scene. We use two cameras on either side of the robot for partial point-cloud of the environment. The camera poses are randomized while maintaining a good view of the whole scene. Additionally, we assume that segmentation masks of the robot and the target object (to fetch) are provided, and the point cloud is converted into the robot-base frame. These assumptions are valid in practice with eye-to-hand calibration, with off-the-shelf segmentation [38], and tracking models [39].

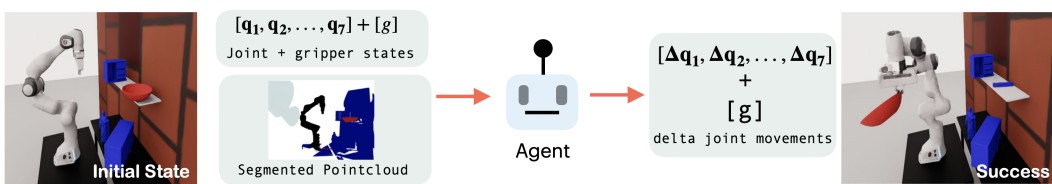

Figure 2: Example of the fetching task, including the initial state, the success grasp state, and the task inputs of segmented point cloud and joint states.

Given the segmented point cloud of the scene and the joint states, the goal of the robot is to fetch the target object from the scene to the free space near the robot (Figure 2). The robot's trajectory is considered successful if the task is completed, with no other object significantly displaced. Further, similar to [13, 12], we use the average computation time and C-space trajectory length of all successful grasps as secondary metrics. Please refer to Appendix A.1 for more details on task setup.

## 4 Benchmark & Dataset

To benchmark the robot's ability to fetch objects in complex environments, we create FetchBench to mimic various commonly occurring but challenging scenarios from the real world (Figure 1). Here, we describe the scene generator and the dataset generation pipeline for fetching.

### 4.1 Procedural Task Generation

**Scenes.** We create 13 types of procedural scenes made with objects such as shelves, cabinets, drawers, dressers, baskets, etc. Each scene type combines several different procedural assets from Infinigen [40, 10]. Our scene generation procedure has a total of 310+ independent parameters that can be randomly sampled to create diverse scene instances. Next, we automatically annotate support surfaces (e.g., table-tops, shelf boards) in procedural scenes to spawn the objects. In particular, there are four categories of support surfaces: on-table, on-shelf, in-drawer, and in-basket. These labels allow us to evaluate methods based on scene category. For more details on procedural scene generation, please refer to Appendix A.2.

**Objects.** While a majority of the objects come from the ACRONYM dataset [41], we additionally include procedurally generated objects from Infinigen [10] such as plates, food bags, containers, forks, etc. We collect a total of 5544 objects and label the grasp poses as in [41, 4] with Isaac-Gym simulator [28]. We use a train-test split of roughly 7:1 for the objects, where the test split is solely used for evaluation and benchmarking purposes.

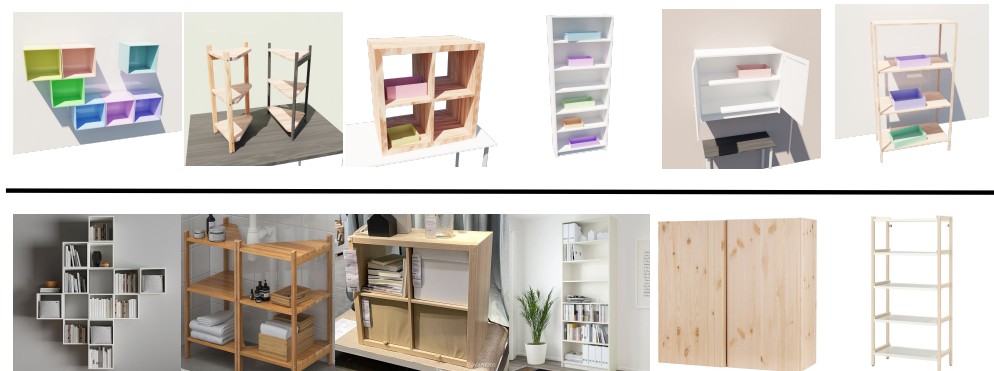

Figure 3: Examples of procedural scenes designed with Infinigen [10] assets (first row). A similar scene can be replicated in the real world with IKEA furniture (second row). The scenes are rendered in Infinigen.

**Tasks.** Each task instance is generated by randomly sampling a scene and placing several objects on the support surfaces with random stable poses, similar to [41]. The robot position, camera poses and the physical constants of friction, coefficient of restitution, and object densities are also randomized within a suitable range.

We note that some task instances might be infeasible to solve, so we filter out instances where (1) an object's initial pose is not static (i.e. the object is falling off after initialization), (2) a collision-free IK solution does not exist for all annotated grasp poses, and (3) the target object is near absent in the input point cloud. In all, we generate a total of 6k tasks for *testing* in our benchmark: 1526 on-table

cases, 2724 on-shelf cases, 891 in-basket cases, and 859 in-drawer cases. Appendix A.3 shows more examples.

## 4.2 Fetch Dataset Generation

To train and evaluate imitation learning methods, we generate our own dataset of fetch trajectories. For a random scene composed of objects in the train split, we iterate over all valid grasp poses of the target object and use motion planning for the approach and retrieval phases. In particular, we use CuRobo with ground truth objects and robot meshes for motion planning. In the retrieving phase, when the object is attached to the end-effector, we use the sample-surface approximation [13] of the target object. We find that CuRobo [13] is more efficient and finds shorter trajectories as compared to OMPL [42] planner.

While unlimited data can be generated with the above pipeline, here we generate a dataset of 27.5k trajectories with over 3.6M frames on 5.7k fetching task instances.

## 5 Baselines

We choose baselines involving both the sense-plan-act pipeline (Figure 4) and the imitation learning approach trained on our Fetch Dataset. For the former, we further consider variants with different motion generation modules: CuRobo [13], RRTConnect [43], and MPPI [44] in Cabinet [9]. For the latter, we consider a fully end-to-end implementation and a hybrid approach that combines the pipeline with the behavior-cloned policy in the retrieval phase.

The sense-plan-act pipeline starts with point cloud inputs, and motion plans the trajectory from the initial state to pre-grasp and similarly from post-grasp to free space. Note that while planning a motion for post-grasp to free space, we treat the object as an additional fixed link to the end-effector.

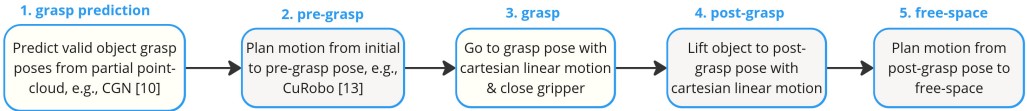

Figure 4: The sense-plan-act pipeline commonly used in grasping frameworks [1].

**ContactGraspNet-[Motion Planning].** We follow the pipeline in Figure 4. Here, we use Contact-GraspNet for 6D grasp pose prediction [1]. We test two motion planning algorithms, i.e., CuRobo [13] and RRT-Connect [43], and name them as CGN-CuRobo, CGN-RRTConnect respectively.

**ContactGraspNet-Cabinet.** Murali et al. [9] proposes a neural collision checker for partially observed objects and scenes. We re-implement the same pipeline as above, but use MPPI [44] iterations with Cabinet's neural collision checking and waypoints proposals as the motion planning module.

**End-to-End Behavior Cloning.** We train an end-to-end policy to grasp the object from the cluster with expert demos. In our architecture, we first encoder the segmented point cloud into 1-D embeddings and concatenate them with joint states and end-effector embeddings. These embeddings are then passed to a backbone to output the delta movements. We use two variants for the backbone: an MLP [19] that takes the current step embeddings, and a transformer [45] that takes in a history of embeddings. We name these E2EImit-MLP and E2EImit-Transformer, respectively. Please refer to Appendix B.2 for details.

**ContactGraspNet-CuRobo-Imitation Learning.** This method is a combination of CGN-CuRobo and imitation learning. It is similar to CGN-CuRobo except for retrieval steps (4) and (5) in Figure 4, where instead we use an end-to-end behavior model to command the robot. Similarly, we implement two variants: CGN-CuRobo-Imit-MLP and CGN-CuRobo-Imit-Transformer.

Table 2: Evaluation of baselines on our benchmark. We show the average success rate, the average computation time and trajectory c-space length on the success cases.

|  | Total Succ | Computation Time (s) | C-Space Length (Rad) |
| --- | --- | --- | --- |
| CGN-CuRobo | 0.094 | 16.4 | 6.17 |
| CGN-RRTConnect | 0.131 | 139.0 | 12.4 |
| CGN-Cabinet | 0.121 | 42.3 | **5.82** |
| E2EImit-MLP | 0.082 | 58.6 | 38.4 |
| E2EImit-Transformer | 0.099 | 57.5 | 22.3 |
| CGN-CuRobo-Imit-MLP | 0.200 | 17.0 | 20.7 |
| CGN-CuRobo-Imit-Transformer | **0.203** | **16.1** | 6.32 |

# 6 Experiments

We perform all experiments in Isaac-Gym [28] simulator on 6k test task instances. We tune all the methods' hyper-parameters on held-out validation tasks generated from the train split.

We design experiments to address the following key questions:

1. How do the baselines compare on FetchBench? Is the fetching task close to be solved?

2. In the pipeline-based approaches, which component(s) present bottlenecks in object fetching?

## 6.1 Baseline Comparisons

Table 2 shows the performance of all baselines on the benchmark. We first note that the maximum success rate that any baseline could achieve is only about 20%, indicating that the task is far from being solved.

Next, we observe that the end-to-end behavior cloned policies have <10% success rate. Specifically, we find that the grasping phase presents significant challenges for an end-to-end trained model, as the model must learn both collision-free planning and generalize to grasp various object shapes. Thus, our dataset and benchmark offer a challenging testbed for researching imitation learning methods in object fetching, where the methods can be evaluated based on their ability to generalize to unknown objects and scenes.

Lastly, we found that the CGN-CuRobo-Imit variants perform the best, with the transformer variant achieving a 20% success rate. Moreover, this variant is nearly as fast and efficient as other motion-planning-based variants. The performance improvement seems to result from strategically decomposing the task into three main components and selecting algorithms optimally suited for each. In the approaching phase, it's more effective to train a grasp pose prediction model that generalizes well to novel objects and scenes rather than an end-to-end policy. During the retrieval phase, the behavior model sidesteps motion planning challenges in partial point-cloud environments by leveraging prior knowledge implicitly learned from training demonstrations.

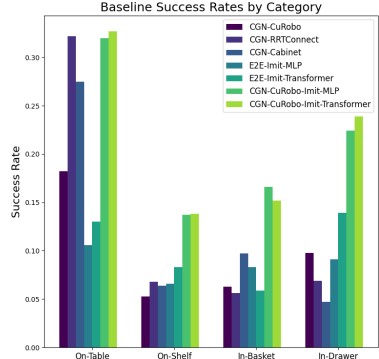

Figure 5: The success rate of different methods by category.

We also compare the performance across the 4 scene categories described in Section 4.1. As shown in Figure 5, fetching from shelf, boards, and baskets is significantly more challenging than from table tops and drawers. This experiment emphasizes the importance of evaluation benchmarks that encompass the diverse scenarios typical in daily life.

## 6.2 Baseline Ablation Study

In this section, we conduct an ablation study for methods that follow the pipeline in Figure 4, specifically testing it with oracle components: ground truth annotated grasp poses and an accurate scene mesh for motion planning context. Here, we answer the following questions:

Table 3: Success rate of different ablations of the common grasping pipeline. Additionally, we break it down into approach phase motion-planning success rate (Approach Plan), approach phase execution success rate (Approach Exec), retrieval phase plan (Retrieval Plan). The approach phase execution is considered successful if the end-effector reaches the commanded pose within a small error threshold.

|  | Approach Plan | Approach Exec | Retrieval Plan | Final Succ |
|---|---|---|---|---|
| GA-Mesh-CuRobo | 0.865 | 0.861 | 0.776 | 0.671 |
| GA-Mesh-RRTConnect | 0.947 | 0.903 | 0.681 | 0.529 |
| GA-Ptd-CuRobo | 0.792 | 0.705 | 0.321 | 0.190 |
| GA-Ptd-RRTConnect | 0.949 | 0.678 | 0.502 | 0.306 |
| CGN-Mesh-CuRobo | 0.611 | 0.602 | 0.546 | 0.336 |
| CGN-Mesh-RRTConnect | 0.607 | - | 0.478 | 0.275 |

**Q1: How successful is the fetching pipeline when it uses accurate grasp annotations and perfect scene meshes?** In this ablation, we replace CGN with annotated grasp poses that offer a collision-free IK solution and employ the ground-truth scene mesh for motion planning. We term this variant as *Grasp Annotation-Mesh Motion Planning* ablation (GA-Mesh-[MP]). In Table 3, we find that even with ideal grasp and scene information, the success rates achieved are only 67% with CuRobo [13] and 53% with RRTConnect [43]. Analyzing the failure cases, we find with GA-Mesh-CuRobo motion planning fails in 14% of cases reaching the pre-grasp pose and 9% reaching the end-state. In another 10% of cases, the object slips from the gripper during fetching. This slipping is often due to the discrepancy between the grasp annotation process in free space and the actual grasp execution in cluttered scenes. While in free space, the object pose adjusts according to the gripper's pose; in cluttered scenes, objects may collide with nearby items or support surfaces, causing failures for several grasp poses.

**Q2: How much does using partial point-cloud data for motion planning contribute to bottlenecks?** We use partial point clouds for motion planning context alongside annotated grasp poses in this ablation, termed GA-Ptd-[MP]. In Table 3, we find that GA-Ptd-CuRobo achieves a 19% success rate, whereas GA-Ptd-RRTConnect reaches 31%. Planning with a partially observed point cloud often tends to underestimate the collisions during execution, leading to more failures: GA-Ptd-RRTConnect finds a collision-free path from the initial state to the pre-grasp pose in 95% of cases but only achieves a 68% success rate in execution. In the retrieval phase, although it plans successfully 50% of the time, the final success rate is just 31%. This drop from planning success to final success is notably larger compared to the GA-Mesh ablation, a trend that is consistent with the CuRobo variant.

**Q3: How much does low-quality grasp pose prediction contribute to bottlenecks?** We use CGN to predict the grasp poses and use scene mesh for motion planning. We denote this ablation as CGN-Mesh-[MP] ablation. We find that CGN-Mesh-CuRobo achieves 34% and CGN-Mesh-RRTConnect achieves 28%, which corresponds to a drop of approximately 50% as compared to GA-Mesh ablations, suggesting space for improvement of the grasp pose prediction model.

**Q4: Can we solve the grasping tasks by naively repeating the pipeline multiple times?** To determine if repeating the fetching procedure can resolve failures, we retry up to five times if it initially fails to complete the task. In Figure 6, we compare between with only one iteration and repeat x5 under different ablations. We observe that repetition improves success rates by a margin of 8%-13% for different ablations. However, the task still remains challenging. In addition, despite the improvements, the extra cost is substantial: the average computation time almost doubled, and

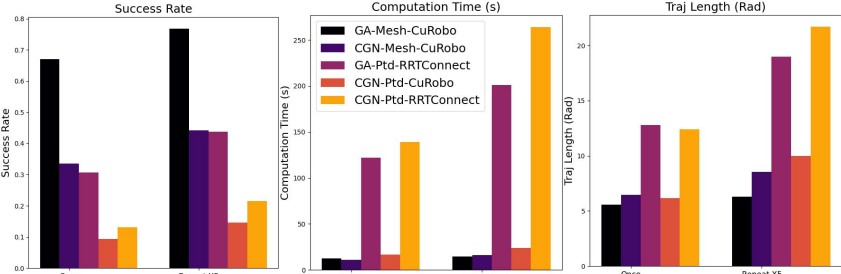

Figure 6: The ablation of repeating the common grasping pipeline by a maximal of 5 times. We show the comparison the success rate, computation time and trajectory length. Here, CGN-Ptd-[MP] denotes standard setting with only point-cloud and joint states as inputs.

the average trajectory length increased by 75%. In all, doing naive retrials is an inefficient strategy, and we believe developing more flexible re-grasping skills is necessary.

## 7   Real Robot Experiment

We show that the challenge of fetching from complex environment exists in real-world environment. Figure 7 shows the example scenes of real-world fetching tasks. We evaluate on 52 on-table tasks, 83 on-shelf tasks, and 32 in-basket tasks with a total of 26 different objects. For the real-robot experiment, we use CGN-RRTConnect with MoveIt![12].

In total, we achieve a success rate of 21%, including 38% on on-table tasks, 13.3% on on-shelf tasks and 12.5% on in-basket tasks. The result suggests the challenge of fetching unknown objects from complex and cluttered scenes (e.g., shelves, bas-

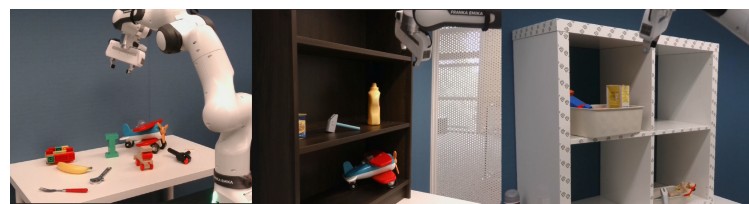

Figure 7: Examples of real-world fetching experiment environments.

kets) in the real-world. Moreover, out of all the failure cases, 25.8 % comes out of low-quality grasp poses (the object slips away), 27.3 % comes out of motion planning failure, 23.5 % comes from unexpected collision with the scene, and the remaining comes from no grasp proposals. We refer to Appendix D for the experiment details, evaluations and videos.

## 8   Discussion and Limitation

In our study, we introduce FetchBench, a simulation benchmark for evaluating various robotic methods for the fetching task. We evaluate several methods on FetchBench, ranging from traditional sense-plan-act pipelines to end-to-end models. These methods struggle in complex scenarios with a maximum success rate of just 20%, indicating substantial room for improvement. Through the ablation studies and the use of oracle components, we have pointed out critical bottlenecks related to partial observability, grasp prediction inaccuracies, and ineffective retrial strategies. We hope that the introduction of this benchmark will serve as a valuable tool for effectively measuring progress in the fetching task, which is a crucial component of many robotic manipulation systems.

Finally, our study has several limitations. First, FetchBench is a simulation-based benchmark that may not accurately represent real-world performance due to sim-to-real discrepancies, especially in physics simulations during contact. Second, our evaluations assume perfectly segmented point clouds, unlike real conditions where segmentation noise exists and non-Lambertian objects present challenges in point cloud extraction. Third, while our framework is general, our evaluations are only centered around Franka arm and gripper. Future work should extend the benchmark to different robotic systems, with increased coverage of object categories.

**Acknowledgments**

This work was partially supported by the National Science Foundation.

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

# A Benchmark Details

## A.1 Task Setup

**Robots and Cameras.** The robot is placed at the front of the scene Figure 9, with +x axis facing forward. The base position is determined by the positions of the support surfaces and the scene type category. For example, for DoubleDoorCabineet type, we place the robot [0.2, 0.5]m beneath the lower deck, to mimic cases when the robot needs to fetch items from the cupboard in the kitchen. The camera positions are also sampled based on the supports. To be specific, we use simple heuristic to set position and look-at position [28], to ensure that the scene objects (shelves, tables) and all surfaces are visible in the field of view. This prevents cases where the robot collide into objects that are completely out of the view. However, we should emphasize that occlusion, e.g., shelf boards, baskets, is still a common challenge and primary cause for failure and collision.

**Initialization.** The robot is initialized with a default joint state. All objects are placed at the pose specified by the configuration file, which is stored after the filtering in Section 4.1.

**Evaluation.** After the algorithm terminates, we wait for an extra 10s for the scene to be stabilized. Then, the grasping is considered successful if the following criterion are satisfied:

1. The center of mass of the target object has a z-value $> 0.3$m in the robot-base frame.
2. The center of mass of the target object has a x-value $< 0.0$m in the robot-base frame. (x+ points the front of the robot)
3. The center of mass of all other objects in the scene has a movement $< 0.1$m from its initialization.

Here, these thresholds are carefully tuned to prevent outlier scenarios.

## A.2 Procedural Scenes

We create 13 types of procedural scenes. Figure 8 shows some examples of the scenes. For the EketShelf, we randomize the size of the cells and its placement on the wall. For the LargeShelf, we randomize the height, width, depth of the shelf, the height of each cell, the basket geometry and size, and the placement of the baskets on each layer of the shelf. For the TriangleShelf, we randomize the board, leg thickness and placement, the board contour geometry, the gap between boards and the

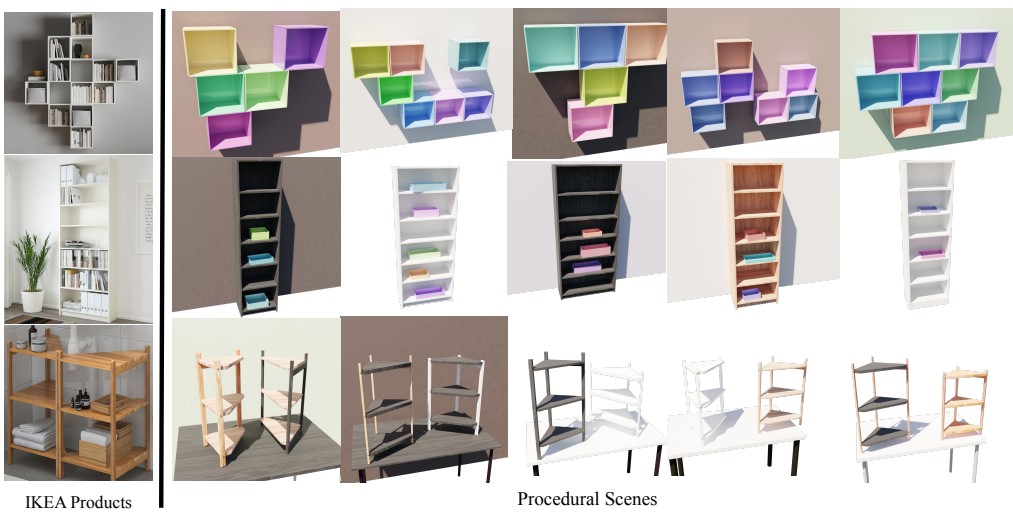

IKEA Products          Procedural Scenes

Figure 8: Examples of procedural scenes and their IKEA counterparts. The procedural scenes are rendered with Infinigen [10]. The first row is the EketShelf type, the second row is the LargeShelf type, and the last row is the TriangleShelf type.

Table 4: Statistics of different types of procedural scenes of the testing tasks.

| | # Scenes | # Tasks | # On-Table | # On-Shelf | # In-Drawer | # In-Basket |
|---|---|---|---|---|---|---|
| CellShelfDesk | 10 | 600 | 265 | 235 | 0 | 100 |
| Desk | 6 | 360 | 360 | 0 | 0 | 0 |
| DeskWall | 7 | 420 | 255 | 165 | 0 | 0 |
| DoubleDoorCabinet | 6 | 360 | 0 | 353 | 0 | 7 |
| Drawer | 6 | 0 | 0 | 0 | 360 | 0 |
| DrawerShelf | 10 | 600 | 0 | 48 | 499 | 53 |
| EketShelf | 5 | 300 | 0 | 300 | 0 | 0 |
| LargeShelf | 10 | 600 | 0 | 481 | 0 | 119 |
| LargeShelfDesk | 9 | 540 | 208 | 266 | 0 | 66 |
| LayerShelf | 10 | 600 | 0 | 362 | 0 | 238 |
| RoundTable | 9 | 540 | 237 | 0 | 0 | 303 |
| SingleDoorCabinetDesk | 7 | 420 | 201 | 214 | 0 | 5 |
| TriangleShelfDesk | 5 | 300 | 0 | 300 | 0 | 0 |
| Total | 100 | 6000 | 1526 | 2724 | 891 | 859 |

size and height of the table-top and the position and rotation of 2 shelves. In Table 4, we show all types of the scenes and statistics in the 6k testing tasks.

### A.3 Task Examples

In Figure 9, we show the examples of our evaluation task from the view of one of the camera, with the Franka robot in place.

## B Baseline Implementations

### B.1 Sense-Plan-Act

For all sense-plan-act methods, i.e, CGN-CuRobo, CGN-RRTConnect, CGN-Cabinet, the CGN takes the point-cloud and the target-object segmentation mask as input, and outputs the candidate grasp poses. The pre-grasp pose is 4cm retracted along the approach direction [33] and the post-grasp pose is 2cm lifted from the grasp pose. These offset values are carefully tuned on held-out validation tasks. For CuRobo [13] and RRTConnect [43] algorithms, the scene point-cloud is converted to mesh with marching cube algorithm with a voxel size of 5mm. For Cabinet [9], the input points to the collision checker and waypoint predictor are first moved to the origin based on the position of the target object. For CGN-CuRobo, we use sample-surface approximation for the partially observed object in the retrieval phase (when it is attached to the end-effector). For CGN-RRTConnect, we use PyBullet and Trimesh for mesh-based collision checking.

### B.2 Imitation Learning

For our imitation learning models, we use PointNet++ [46] and Voxel-CNN [47] as the encoder on segmented point cloud of the target object, the scene and the robot. Furthermore, we encode the proprioceptive states with a MLP encoder. We concatenate all embeddings as the final embedding to feed into the policy network. For the MLP variant, we use the embedding of current step and use a 3-layer MLP to output the delta movement. The output is parameterized as a Tanh Gaussian distribution. For the Transformer variant, we use the consecutive last 4 steps' embeddings to feed into an Optimus policy head [45] and output a Tanh Gaussian mixture distribution.

Additionally, as expert trajectories are collected in two stages, i.e., approaching and retrieval, we augment the model input with one additional one-hot vector to indicate the stage during training. Furthermore, we require the model to output one additional signal of whether the stage should come to an end (termination output). Namely, for all the training data, the very last step of each trajectory stage will have value 1 otherwise −1.

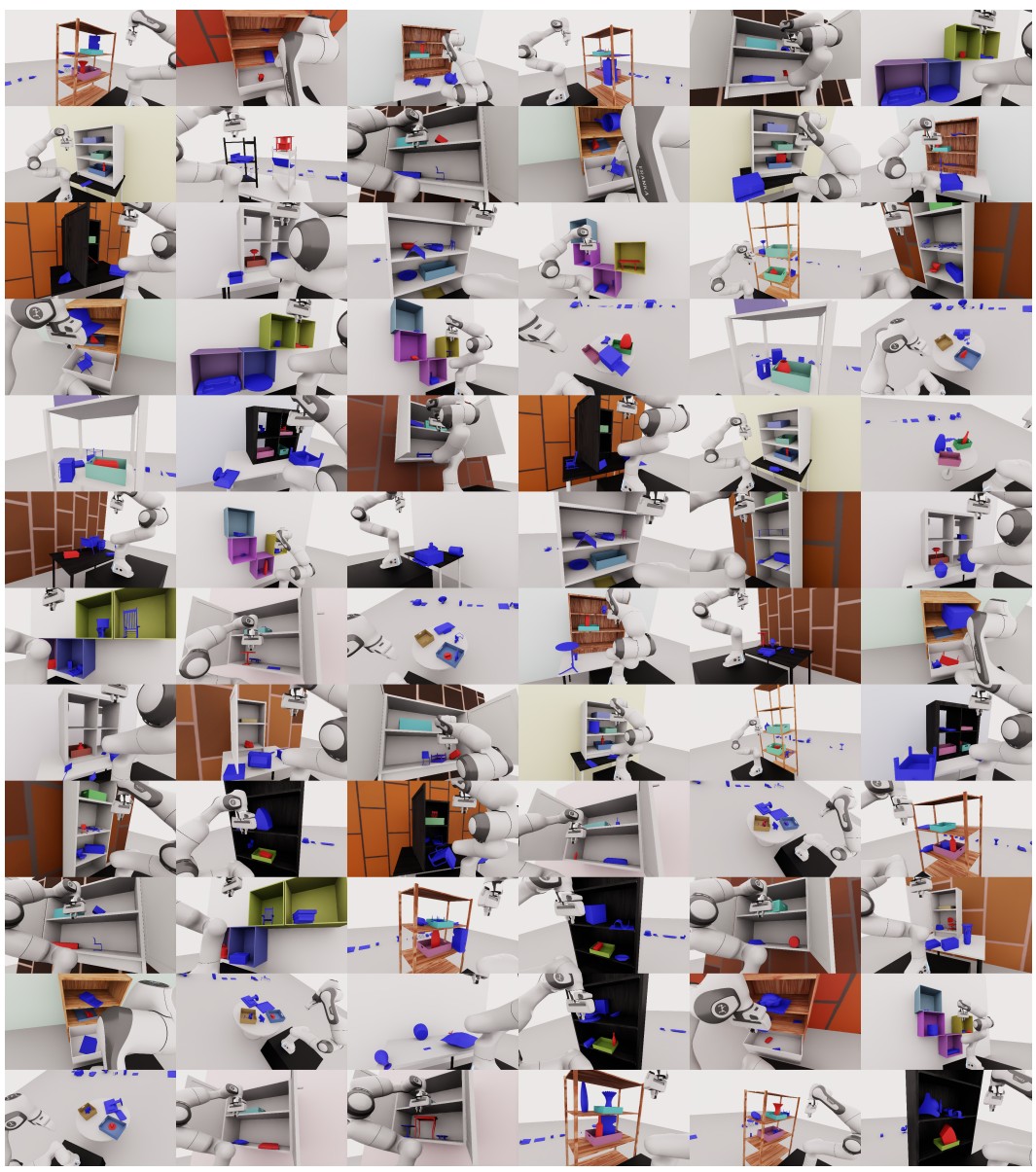

Figure 9: Examples of grasping tasks from one of the camera view when the robot is in place. The scenes are rendered with Isaac-Sim [8].

Though the model is trained for different stages, for E2EImit variants, we use the model in an end-to-end mode with re-grasping behavior. To be specific, if the model is in approaching stage and it outputs the termination signal, then it transits to the retrieval stage and close the gripper. However, if the gripper is fully closed during the retrieval stage (we know the grasp is a failure), then we open the gripper and transit back to the approaching stage. For the CGN-CuRobo-Imit methods, we only use it in the retrieval stage.

All models are trained on our demonstration dataset with Adam [48] optimizer. We select the best hyper-parameters by validating on held-out validation tasks. In Appendix C, we show more results on our imitation learning models.

# C  Additional Benchmark Experiment

We provide additional experimental results on FetchBench.

## C.1  Other Manipulators

Our benchmark uses Franka Research 3 robot. However, we should note that the benchmark environments can be easily extended to other robots.

Different robot hardware will affect the overall performance of the fetching task. In Table 5, we evaluate two different hardware setups: Kinova Gen3 with Franka Hand and Kinova Gen 3 with Robotiq Hand. For simplicity, we use the ablation experiments of CGN-CuRobo.

We found that altering the arm morphology alone only slightly impacts performance due to variations in motion planning problems. However, changing the gripper significantly affects performance. This is because grasp annotations were evaluated using the Franka Hand [41], while the Robotiq gripper has a different morphology and different physical process when closing the gripper.

Table 5: Results of success rate in ablation experiments on different robot hardwares.

|  | Franka Arm Franka Hand | Kinova Gen3 Franka Hand | Kinova Gen3 Robotiq Hand |
|---|---|---|---|
| GA-Mesh-Curobo | 0.671 | 0.629 | 0.471 |
| GA-Ptd-Curobo | 0.190 | 0.173 | 0.108 |
| CGN-Curobo | 0.094 | 0.088 | 0.017 |

## C.2  Imitation Learning Baselines

We explore to improve the performance of our imitation learning baselines. First, we show that the end-to-end deployment with re-grasping behavior does improve the performance by roughly $3\%$ in terms of success rate, over the vanilla two-stage deployment (i.e., only approach-then-retrieval). However, we also found that the re-grasping behavior also increases the computation time and c-space trajectory length significantly. The results of E2EImit variants and TwoStageImit variants are shown in Table 6.

Table 6: Results of imitation learning baselines with different inference mode.

|  | Success | Computation Time | C-Space Trajectory |
|---|---|---|---|
| E2EImit-MLP | 0.082 | 58.59 | 38.42 |
| TwoStageImit-MLP | 0.053 | 16.54 | 20.88 |
| E2EImit-Transformer | 0.099 | 57.7 | 45.1 |
| TwoStageImit-Transformer | 0.065 | 16.67 | 10.30 |

In addition, we also evaluate the performance with twice amount of training data and with ACT [49] algorithm. The results are shown in Table 7. Training with twice amount data and on more advanced algorithms do improve the performance. We leave it as future work to improve the end-to-end imitation learning models on challenging fetching tasks.

Table 7: Results of imitation learning methods under different models and with more training data.

|  | Success (1x Data) | Success (2x Data) |
|---|---|---|
| TwoStageImit-MLP | 0.058 | 0.061 |
| E2EImit-MLP | 0.082 | 0.093 |
| E2EImit-ACT | 0.093 | - |

# D  Real Robot Experiment

## D.1  Experiment Setup

We demonstrate that robot fetching from complex environment is a challenging problem in practice. Here, we implement the CGN-RRTConnect baseline and test on a diverse set of objects and environments in order to compare performance between different scenes.

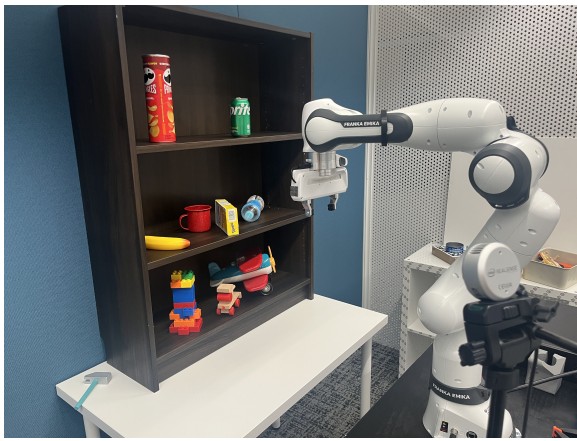

Figure 10: Hardware setup of our real world experiment. The Franka Emika Research 3 is placed in front of the scene and we use Realsense L515 pointing towards the scene to capture RGB-D information.

**Hardware and Evaluation Setup.** For our hardware, we used the 7-DOF Franka Emika Research 3 as our robot arm, and a Intel Realsense L515 LiDAR camera to capture the RGB-D images of the scene [1]. In order to evaluate on a diverse set of environments, we performed experiments on a tabletop, two distinct shelves and various baskets. Figure 10 shows our hardware setup and the example scenes are shown in Figure 7.

We tested on a diverse set of objects in our experiment. Figure 11 shows examples of the objects used in our experiments.

**Algorithm.** With our real-world robot, the CGN-RRTConnect baseline is implemented as follows.

1. Capture the RGB-D image of the scene and get the target object segmentation mask by prompting Seg-Any.
2. Use the point-cloud and the mask to query CGN for candidate grasp poses on the target object.
3. Use MoveIt! [12] to search for motion to the pre-grasp poses based on the confidence scores.
4. Move from pre-grasp pose to grasp pose with linear motion from pilz's LIN planner [12].
5. Crop out the object from the point-cloud, add a placeholder to the robot, and plan the motion to the initial pose.

## D.2  Experiment Results

Table 8 shows the detailed results of our real-world experiment. Comparing between table-top and shelf cases, we see the % success significantly decreases in shelf scenes due to the increased difficulty of the environment. In addition, we also show that grabbing objects from baskets within each respective scene also results in lower % success due to physical constraints.

**Failure Analysis.** To further understand the behavior of the CGN-RRTConnect baseline, we broke down each unsuccessful attempt into four categories of failure:

1. No Grasp Poses (NGP), where CGN failed to find any grasps.
2. Motion Planning Failure (MPF), where CGN returned grasps, but RRT-Connect found no valid trajectories.

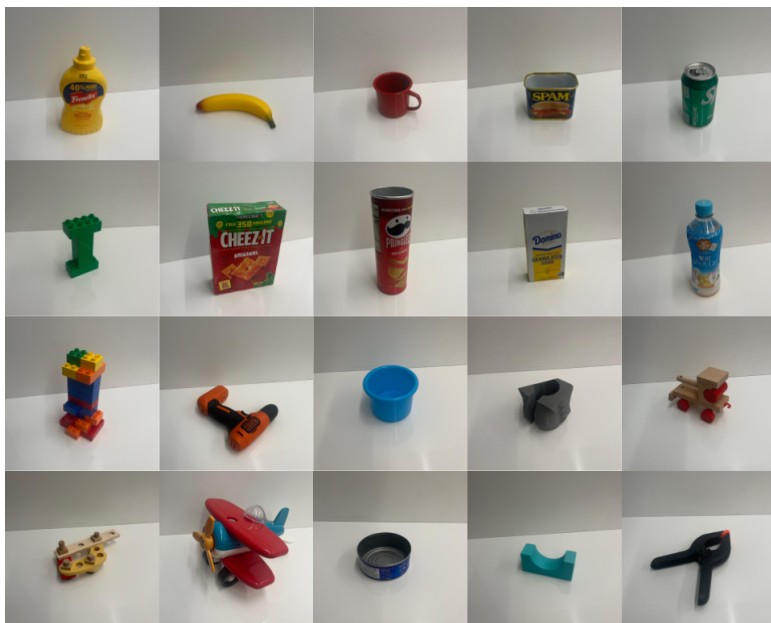

Figure 11: Examples of objects used in our real world experiments.

|  | # Success | # Attempts | % Success |
|---|---|---|---|
| TableTop | 20 | 52 | 38.5 % |
| TableTop-Basket | 3 | 16 | 18.8 % |
| Shelf | 11 | 83 | 13.3 % |
| Shelf-Basket | 1 | 16 | 6.25% |
| Total | 35 | 167 | 21.0 % |

Table 8: Results of real-world fetching experiment by environment types.

3. Invalid Grasp Pose (INV), where the robot attempts to grab the object, but the grasp is unstable.
4. Collision Failure (CF), where the robot or target object collides/disrupts the environment.

|  | % Success | % NGP | % MPF | % INV | % CF |
|---|---|---|---|---|---|
| TableTop | 38.5 | 13.5 | 0 | 36.5 | 11.5 |
| TableTop-Basket | 18.8 | 31.3 | 18.8 | 25.0 | 6.25 |
| Shelf | 13.3 | 15.7 | 31.3 | 12.1 | 27.7 |
| Shelf-Basket | 6.25 | 37.5 | 43.8 | 6.3 | 6.3 |
| Total | 21.0 | 18.6 | 21.6 | 20.4 | 18.6 |

Table 9: Results of different failure types in each type of environment.

Table 9 shows the failure types of all attempts in each environment category. Comparing table top scenes to shelf scenes, shelves have a much higher failure rate due to a higher number of motion planning and collisions. This is attributed to the difficulty of motion planning within a shelf scene. In addition, basket scenes had a higher rate of NGP failure compared to their normal counterparts, as the baskets produce occlusions that limit the pointcloud input into CGN. We also found that many of the perspectives in shelf and basket scenes caused CGN to output very few + low confidence grasp propositions, leading us to believe that CGN is sensitive to perspective and produces better results on table top scenes.

**Limitations.** Due to limited resources, we can only evaluate on a smaller number of fetching cases, comparing to the simulation benchmark.

# E  Additional Related Works

**Grasp Pose Prediction.** Predicting grasp poses for various novel objects has been an important long-standing research challenge in robotics [50]. It becomes more challenging to predict accurate and diverse grasp poses from noisy and partially observed sensory readings like depth maps and partial point clouds [1, 5]. Recently, learning-based grasp pose synthesis has become a crucial solution paradigm to this problem [33], owing to the power of neural networks to handle high-dimensional inputs and flexibility to various objects with different geometries and physical properties. To train the neural network, methods create and utilize large-scale object grasp pose datasets [41, 2, 51]. The valid grasp poses of each object are labeled with analytic metrics [3, 5, 2], or with physics simulation [35, 1]. Notably, researchers have applied the grasp pose prediction models to build robust grasping systems that can clear various unknown objects from cluttered bins [5] and table-tops [1].

Grasp pose prediction models play a crucial role in the baselines we tested. We use the Contact-GraspNet [1] to predict 6D grasp poses from partial point clouds to command the motion generation module. However, as we will show in Section 6.2, the model is not powerful enough to solve the benchmark. Furthermore, having an accurate grasp pose is not the only bottleneck challenge to grasping objects from more challenging cluttered environments, e.g., shelves, cabinets, and drawers, that are crucial for applications like service robots [7].

**Motion Generation for Robot Arm.** To generate collision-free trajectories is one of the fundamental problems for robot arm control. Given the obstacles of the environment, sampling-based motion planning algorithms like RRT [52] and its variants [43] are the common choice to command the arm to the target pose [12]. Meanwhile, optimization-based motion generation methods [13] have been proposed as an alternative to the classical approach. CuRobo [13] has much higher efficiency than the sampling-based motion planning algorithms, as it can be computed in parallel on GPUs. However, these methods assume a known environment and obstacles. They do not account for the partial observation problem, which is a common challenge for in-the-wild applications like home robots. To overcome this issue, [9, 53] propose to learn a neural collision checker for partially observed scenes and movable objects. After training on large-scale synthetic data, these models have shown promising results in tackling the partial observation problem in robot object rearrangement tasks.

However, in Section 6.2, we find that the partial observation problem still remains a challenge to existing motion generation methods in grasping. This suggests huge space for improvement in the motion generation methods to tackle real-world challenges.

**Imitation Learning.** Imitation learning [49, 54, 45] has become a promising approach to learning large-scale behavior models [36, 37] for robots. However, despite great effort from the community, researchers still lack enough data to train powerful large behavior models. Moreover, a majority of the expert data [36, 55, 56, 57] are collected on table-top environments and lack diversity in scene variation. With our procedural scenes and tasks, our simulation benchmark can generate a large quantity of diverse grasping demonstrations in various environments. Thus, our benchmark also serves as a platform for imitation learning research of large behavior models on grasping. We provide a procedural demonstration synthetic data generator for diverse and abundant expert demos. Furthermore, in Section 6.1, we find that combining imitation learning and the common grasping pipeline achieves SOTA performance on our benchmark, which suggests a promising direction for new grasping systems.

