# OpenReview forum: "FetchBench: A Simulation Benchmark for Robot Fetching"
_robot-learning.org/CoRL/2024/Conference — CoRL 2024_

### Official Review · Reviewer_vPSQ · 2024-07-15
**Reviews on the FetchBench**

**Originality:** 3
**Technical Quality:** 3
**Clarity Of Presentation:** 3
**Potential Impact:** 3
**Recommendation:** 2
**Confidence:** 4

**Review:**

Strengths:

- The design of FetchBench, including the data generation pipeline is well-thought-out.
- The empirical analysis revealing a maximum success rate of only 20% is insightful.
- The procedural generation of scenes and tasks is a good approach to creating diverse and unlimited test cases.
- The baseline method is also provided along with the benchmark.


Limitations:
- Currently, only one robotic hardware is provided, which greatly limits the general usage of the benchmark.
- This benchmark focuses more on the robotic fetch tasks, which is the one important yet partial process of the whole robotic manipulation tasks.
- The evaluations are conducted with the perfectly segmented point could which is obviously impossible in the real-world application. Meanwhile, the perception result should affect the manipulation tasks quite huge.
- Currently, only one type of gripper is provided.
- Some details on the assets, e.g. target object is missing in the paper. Something like how the physical materials, whether they fragile or not, etc.

**Quality Of The Limitations Section:**

3

**Questions For Rebuttal:**

- See the reviews section.
- How the material or other physical properties are distributed in the benchmark.
- Can the robotic arm adjust its pose before the fetch, or it is the fixed base?
- Would different robotic hardware affect?
- When spawning the robotic arm, how to consider the action space of the robotic arm?

**Robotics Focus:**

3

**Summary Of Paper:**

This paper proposes a new benchmark FetchBench, with diverse procedural scenes that integrate both grasping and motion planning challenges.

**Summary Of Recommendation:**

More details are needed and the benchmark has the potential to get scaled and diversified.

---

### Official Review · Reviewer_qTGA · 2024-07-17
**Interesting benchmark to stress-test analytical and learning based methods for pick and place manipulation**

**Originality:** 3
**Technical Quality:** 3
**Clarity Of Presentation:** 3
**Potential Impact:** 3
**Recommendation:** 3
**Confidence:** 3

**Review:**

Strengths:
* The paper is generally well-written. The authors clearly define the problem of existing benchmarks, describe (most) related work and list their limitaitons.
* The experiments and ablations are thorough, comparing state-of-the-art approaches, and providing a detailed analysis of the specific failure modes of each approach
* The implications of the study are informative and somewhat surprising. The results suggest that neither analytical nor learned-based grasping methods are performant and that significant work remains to improve their ability to generalize to novel scenes and objects.

Areas for improvement:
* The amount of generated demonstrations to train the imitation learning baseline is relatively small (27.5k), given the large variety of objects and procedurally generated scenes in this benchmark. It would be interesting to see if increasing the number of demonstrations one or two orders of magnitude would help to address the limitations of the imitation learning baseline. Increasing the magnitude of data in this setting should be relatively cheap, as this is done in simulation autonomously.
* The experiments indicate that a hybrid approach that uses analytical grasping techniques and imitation learning produces the best results. Beyond this conclusion, the paper is missing a discussion on the broader implications. Is the implication that grasping based techniques will suffer from bottlenecks, and should be phased out? Are there any suggestions for how these bottlenecks can be addressed? Furthermore, for pure imitation learning approaches, what do the authors suggest to improve their performance? Increasing data quantity, improving model capacity, etc?
* the focus on pick and place tasks limits the scope of the benchmark. Is it possible to extend it to also include articulated manipulation, for example opening and closing cabinet doors, and drawers?

**Quality Of The Limitations Section:**

3

**Questions For Rebuttal:**

Please see the points for “areas for improvement” in the previous section. Furthermore, please add a discussion on RoboCasa [1] on the discussion on related work, which also includes pick and place tasks and imitation learning datasets for these tasks.

[1] Nasiriany et al., RoboCasa: Large-Scale Simulation of Everyday Tasks for Generalist Robots. RSS 2024

**Robotics Focus:**

4

**Summary Of Paper:**

The authors present FetchBench, a simulation benchmark focused on diverse pick and place manipulation tasks. The benchmark consists of a set of procedurally generated scenes with 5.5k+ objects, a dataset of 27.5k successful grasp trajectories, and a study comparing analytical, learned, and hybrid grasping models on these tasks. The study shows that all of these approaches have significant room for improvement. The best approach is a hybrid approach which uses analytical grasping methods to reach and grasp objects, and imitation learning methods to manipulate the object post-grasping. A series of more fine-grained analyses identify specific bottleneck parts where each method struggles, which is insightful.

**Summary Of Recommendation:**

It is insightful to see that, contrary to a commonly held belief that pick and place tasks are “solved”, existing state-of-the-art approaches fail to solve these tasks effectively. I believe that this benchmark will be helpful to the community to help develop more effective approaches.

---

### Official Review · Reviewer_XzpA · 2024-07-18
**A simulated robotic fetching benchmark in static but cluttered environments**

**Originality:** 3
**Technical Quality:** 3
**Clarity Of Presentation:** 4
**Potential Impact:** 3
**Recommendation:** 3
**Confidence:** 4

**Review:**

### Strengths
* The paper is well organized and clearly written.
* The proposed benchmark provides procedurally generated scenes, that can effectively evaluate the generalization capability of a robot manipulation policy, which hasn't been available but realistic and important. Thus, it is a good contribution to the community.
* The benchmark and benchmarking protocol is clearly explained and makes sense.
* The benchmarking experiments and ablation studies pinpoint the difficulties in grasp pose estimation and generating motions under partial observabiilty.

### Weaknesses
* The imitation learning baselines seem not properly evaluated to reach the conclusion or it requires more experimental backups (especially with several state-of-the-art algorithms and 2D/3D observation options). Although the "generalization" aspect of the proposed benchmark could be very challenging for such learning-based approaches, given how the latest imitation learning methods perform well in diverse manipulation task setups, these SOTA imitation learning methods are expected to achieve some meaningful results.
* (optional) it would be very interesting to see how reinforcement learning approaches perform on this benchmark.

**Quality Of The Limitations Section:**

3

**Questions For Rebuttal:**

* The end-to-end imitation learning baselines seem too weak (less than 2% success rates). The authors could consider some latest approaches, such as Diffusion Policy, ACT, and OpenVLA. Moreover, the ablation studies are only conducted for planning-based approaches; but, the authors could consider including ablation studies for imitation learning approaches to provide more insights about the benchmark tasks for end-to-end learning.
* In Table 2, "Traj Length (Rad)" is a bit confusing to me. It would be clear to use "Traj Length in C-Space" instead. Moreover, it would be interesting to see the "Traj Length" in terms of execution time or number of low-level control actions, which might be something around 600 (3M / 5K).

**Robotics Focus:**

3

**Summary Of Paper:**

This paper proposes a new simulated benchmark for robotic fetching tasks in cluttered environments, which requires a Franka robot arm to grasp a desired object and fetch it in diverse scenes, including a table, shelf, drawer, and basket. Despite the abundant prior simulated robotic manipulation benchmarks, this benchmark poses its unique challenges and anlyses: (1) procedurally generated 6K testing task instances to evaluate a policy's **generalization capability** to diverse scenes and objects, (2) dissecting a "sense-plan-act" pipeline to find out its major roadblocks, which are mainly due to *partial observability* and *grasp pose estimation*. This simulated benchmark can serve as a testbed for generalizable robotic manipulation.

**Summary Of Recommendation:**

This benchmark provides an important testbed for "generalization", which has been barely covered in the robotic manipulation community. Thus, I believe this benchmark will be a good contribution to further robot learning research. I suggest "weak acceptance" of this paper due to relatively weak imitation learning baselines, which may be able to show reasonably good performance in this benchmark.

---

### Author Rebuttal · Authors · 2024-08-13

We thank all reviewers and the area chair for their time and insightful comments and feedback. We are pleased that the reviewers found the paper:

(a) **to have a potential impact.** Reviewers (XzpA, qTGA) state that the benchmark will be helpful to the community in developing effective approaches.

(b) **thoughtful benchmark design.** the benchmark evaluates robot fetching at massive scale with procedural assets. (vPSQ)

(b) **well-organized and clear,** addressing the limitations of prior works (XzpA, qTGA)

(c) **includes “informative and insightful study,” “thorough experiments and ablations”,"somewhat surprising results"** and analysis of specific failure modes, and pinpoints existing challenges in fetching (XzpA, qTGA, vPSQ)

We also thank the reviewers for raising thought-provoking questions and concerns, which we address as individual responses to the reviewers.

---

### Decision · Program_Chairs · 2024-09-04

**Decision:**

Accept

**Comment:**

This is a new simulation benchmark for fetching. The authors added new experiments during the rebuttal, with results now using ACT and Diffusion Policy. The authors also did new experiments with different data sizes for imitation learning, as well as changing the robot type and gripper. There was some internal discussion about this benchmark among reviewers during the rebuttal. While a unanimous accept did not emerge from this discussion (especially due to potentially unrealistic assumptions like perfectly segmented point clouds), the consensus was that there is merit to the benchmark. It appears a good fit for the robot learning community, though there is overlap with some other benchmarks such as RoboCasa (https://robocasa.ai/).

The authors should add their new rebuttal experiments to the finalized paper.

Strengths:
- The paper studies a novel application area for a benchmark.
- While it is in simulation, which makes it easier to compare different algorithms from different research teams, it shows applicability in the real world.
- The design of the benchmark is sound overall, with procedural generation involved to test generalization. Also, it tests occlusions.

Weaknesses:
- The imitation learning results could probably be better than they are reported in the paper. For example, how about increasing the size of the demonstration data?
- The paper could also consider adding more imitation learning (and reinforcement learning) methods.
- It would be nice to have more focus on diverse robots and grippers.